# Task Priors: Enhancing Model Evaluation by Considering the Entire Space of Downstream Tasks

## Abstract

The grand goal of AI research, and particularly Self Supervised Learning (SSL), is to produce systems that can successfully solve *any* possible task. In contrast, current evaluation methods available to AI researchers typically rely on a fixed collection of hand-picked downstream benchmarks. Hence, a large amount of effort is put into designing and searching for large collections of evaluation tasks that can serve as a proxy for our grand goal. We argue that such a rigid evaluation protocol creates a silent bottleneck in AI research. To remedy that, we define a *probabilistic space of downstream tasks* obtained by adopting a distribution over tasks and by defining **Task Priors** Under this view, one can evaluate a model's performance over the set of all possible downstream tasks. Beyond establishing a new standard for evaluation, we believe that **Task Priors** will accelerate the pace of research in SSL–where downstream task evaluation is generally the sole signal that researchers have access to.

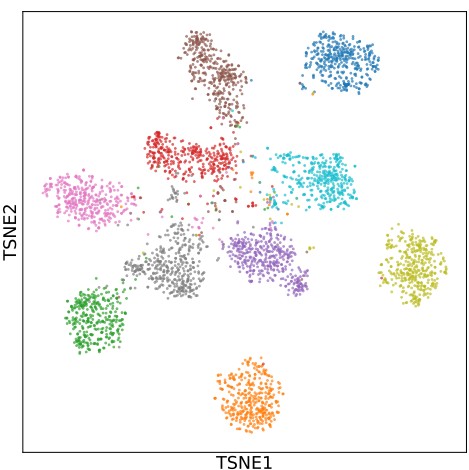

(a) TSNE of Imagenette [5] with standard labels.

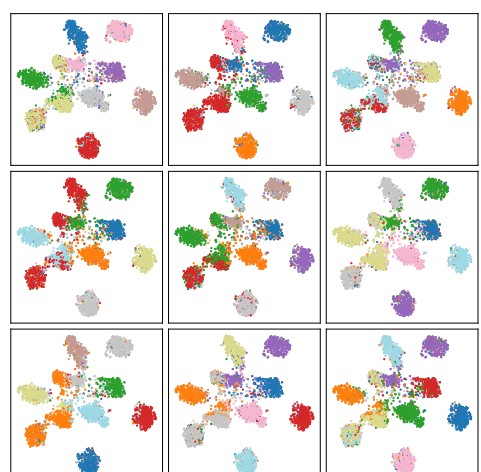

(b) Labels Generated by the *Task Prior*.[1]

Figure 1: Comparison of the naive way to evaluate a model, only on the specific choice of labels provided with the dataset (Left) with the probabilistic view of targets generated by sampling from the Task Prior, giving us a distribution we can evaluate on (Right). In our Task Priors framework, we can evaluate on an infinite space of downstream tasks.

# 1 Introduction

Pretrained backbone models today are released as a single checkpoint, yet in practice they can power millions of distinct downstream tasks: from simple classification and retrieval services to large-scale recommendation, autonomous perception, and more. On HuggingFace alone, top models such as *mobilenetv3*, *clip-vit*, and *bert* can receive over 100 million downloads per month [18]. As the number of users and the diversity of applications grows, the space of possible downstream tasks users want their models to perform well on tends toward infinity.

Yet our standard evaluation protocols remain tethered to a small, fixed suite of hand-picked benchmarks—often half a dozen or so datasets (e.g., ImageNet, COCO, GLUE, SuperGLUE, WMT) [6, 13, 10, 17]. Each new benchmark can take months of expert labeling and tens or even hundreds of thousands of dollars to assemble. Even large-scale benchmark suites that aggregate many tasks — such as the Massive Text Embedding Benchmark (MTEB), which spans 56 different evaluation datasets, or the similar Massive Image Embedding Benchmark (MIEB) — still represent only a narrow slice of the possible task space [12, 19]. Once built, a static benchmark can only ever probe a tiny corner of the real-world tasks for which a model might be deployed. This disconnect creates a structural bottleneck between the handful of evaluation suites we all agree to evaluate our models on and the effectively infinite variety of tasks practitioners use our models for.

One way to break this bottleneck is simply to keep spending more time and money on ever-larger benchmarks, but that approach quickly becomes unsustainable. Instead, we propose a surrogate evaluation framework that treats downstream tasks as samples from a well-defined probabilistic space. By adopting a "Task Prior", a distribution over all possible targets informed by a pretrained feature kernel, we can compute expectations and variances of a model's downstream performance in closed form, without training new classifiers or designing new benchmarks.

# 2 Task Priors

## 2.1 A Distribution Over Target Labels

Given our data $\mathbf{X}$, we will define the one-hot matrix of class labels as $\mathbf{Y}$. Now we can define a graph, where two data points are connected when they are in the same class. This graph would have adjacency matrix $\mathbf{G} = \mathbf{Y}\mathbf{Y}^\top$. We would like to introduce a prior distribution over all such label graphs $\mathbf{G}$ that reflects the likelihood of different downstream tasks.

We define a measure over all possible adjacency matrices, weighted by alignment with a pretrained feature kernel $\mathbf{K}$. This *Task Prior* allows us to compute expectations and variances of kernel alignment scores in closed form, or to efficiently sample realistic downstream targets without training additional classifiers. This will allow us to look into the average performance of our model, but also have the ability to look into other statistical properties.

Suppose that we have a kernel function that measures similarity between data points $k : \mathbb{R}^D \times \mathbb{R}^D \to \mathbb{R}$. Let $\mathbf{K}$ then be a kernel matrix corresponding to $\mathbf{X}$. For the rest of the paper we can assume this is the centered kernel matrix corresponding to the features. Now if we have a graph adjacency matrix $\mathbf{G}$, we can read off the elements of the product, $[\mathbf{GK}]_{ij} = \sum_{k=1}^N \mathbf{G}_{ik} \, k(\mathbf{x}_j, \mathbf{x}_k)$. In particular we can see that the diagonal elements of this matrix are given as follows, where we use $\mathbf{x}_j \sim \mathbf{x}_i$ to mean that $\mathbf{x}_j$ and $\mathbf{x}_i$ are connected in graph $\mathbf{G}$,

$$[\mathbf{GK}]_{ii} = \sum_{k=1}^N \mathbf{G}_{ik} \, k(\mathbf{x}_i, \mathbf{x}_k) = \sum_{\mathbf{x}_k \sim \mathbf{x}_i} k(\mathbf{x}_k, \mathbf{x}_i).$$

Summing over $i$ gives the trace, $\mathrm{Tr}(\mathbf{GK}) = \sum_{i=1}^N \sum_{\mathbf{x}_k \sim \mathbf{x}_i} k(\mathbf{x}_i, \mathbf{x}_k)$, which acts as a global *alignment score* between the label graph $\mathbf{G}$ and the kernel $\mathbf{K}$. We can treat the negative trace, $\mathcal{E}(\mathbf{G}) := -\mathrm{Tr}(\mathbf{GK})$, as the "energy" of a labeling: graphs that connect feature-similar points (high trace of $\mathbf{GK}$) have lower energy and are therefore more likely. This leads naturally to the Gibbs measure

$$\mu(\mathbf{G}) \; \propto \; e^{-\mathcal{E}(\mathbf{G})/T} \; = \; e^{\mathrm{Tr}(\mathbf{GK})/T},$$

---

[1]Interested readers can experiment with different class counts and temperature settings in this Colab notebook: https://colab.research.google.com/drive/1qNOgoNSH87AcdODug-yop7Q0MuT8w1r7

with temperature $T > 0$. As we increase the temperature, this distribution tends towards the uniform distribution on all graphs, and we get more interesting behavior in the low temperature regime. The properties of this distribution enable direction computation of the expectation and higher moments, and enable efficient sampling algorithms which we develop in the sequel.

**Definition 2.1.** *(Task Prior Distribution) Given a kernel matrix $\mathbf{K}$ on $n$ data points, and a temperature $T > 0$, we will define the following Gibbs measure on the space of all graphs, G as:*

$$\mu(\mathbf{G}) \propto e^{\frac{\mathrm{Tr}(\mathbf{GK})}{T}}, \tag{1}$$

*where we denote by $Z_{T,\mathbf{K}}$ the corresponding partition function.*

Although computing exactly the probability of observing a single graph can be quite challenging, as computing the partition function would require $2^{N^2}$ computations, the specific structure of this probability measure admits a neat factorization on a per-edge level.

**Lemma 2.2.** *Suppose that we consider the Gibbs measure over all graphs $\mathbf{G}$. Then, the probability of a single edge $i, j$ being present is given by,*

$$\mathbb{P}(\mathbf{G}_{i,j} = 1) = \sigma(\mathbf{K}_{i,j}/T), \tag{2}$$

*where $\sigma$ denotes the sigmoid function. Furthermore if $(i, j) \neq (l, k)$, then,*

$$\mathbb{P}(\mathbf{G}_{i,j} = 1 \;\wedge\; \mathbf{G}_{i,j} = 1) = \sigma(\mathbf{K}_{i,j}/T)\sigma(\mathbf{K}_{l,k}/T). \tag{3}$$

The above lemma allows us to, given some kernel matrix driven by an assumption on similarity over our data points, $K$, evaluate the performance of a representation model providing another kernel matrix $M$.

**Theorem 2.3.** *Given a kernel matrix $\mathbf{K}$ and associated Gibbs measure $\mu_{\mathbf{K}}$, and another kernel matrix $\mathbf{M}$, we can compute the expectation of $\mathrm{Tr}(\mathbf{MG})$ as follows,*

$$\mathbb{E}_{\mathbf{G} \sim \mu_{\mathbf{K}}} \left[ \mathrm{Tr}(\mathbf{MG}) \right] = \sum_{1 \leq i,j \leq n} \mathbf{M}_{i,j} \mathbb{P}_{\mathbf{G} \sim \mu_{\mathbf{K}}}(\mathbf{G}_{i,j} = 1) = \sum_{1 \leq i,j \leq n} \mathbf{M}_{i,j} \, \sigma(\mathbf{K}_{i,j}/T). \tag{4}$$

*Furthermore, the variance satisfies,*

$$\mathrm{Var}(\mathrm{Tr}(\mathbf{MG})) = \sum_{1 \leq i,j \leq n} \mathbf{M}_{i,j}^2 \, \sigma(\mathbf{K}_{i,j}/T)(1 - \sigma(\mathbf{K}_{i,j}/T)).$$

We will note that computing the mean and variance of $\mathrm{Tr}(\mathbf{MG})$, when $\mathbf{G}$ is distributed according to the Task Prior, takes on the order of $O(N^2)$ computations for $N$ data points. In practice, computing a model's mean and variance now will take only milliseconds.

### 2.2 Empirically Sampling Tasks for Evaluation

Starting from the task-prior distribution $\mu_{\mathbf{K}}$ over graphs introduced in the previous section, we can view every edge as an independent Bernoulli variable whose success probability is $\sigma(\mathbf{K}_{i,j}/T)$. However, for the purpose of measuring performance of a model with linear probes, we may instead want to sample from the probability measure restricted on those graphs $\mathbf{G}$ which arise from one hot labelings $\mathbf{Y}$, where $\mathbf{G} = \mathbf{Y}\mathbf{Y}^\top$. We will denote by $\mu_{\mathbf{K}}^q$ the probability measure given by,

$$\mu_{\mathbf{K}}^q(G) \propto \begin{cases} \mu_{\mathbf{K}}(\mathbf{G}) & \text{if } \mathbf{G} = \mathbf{Y}\mathbf{Y}^\top \text{ for some one-hot } Y \in \{0,1\}^{N \times q} \\ 0 & \text{else} \end{cases}.$$

Sampling from the restricted measure $\mu_K^q$ is a much more challenging problem. For binary labelings there are $2^n$ possible states, so sampling and computing the partition function can be completely intractable. We could use Markov Chain Monte Carlo (MCMC) methods such as the Metropolis-Hastings algorithm to sample from this distribution, but this can also prove to be challenging in practice. Instead, we propose an approximate sampling algorithm in $O(n)$ time to sample a labeling on $n$ data points.

Suppose we write the labeling $\mathbf{Y} = [y_1, \ldots y_n]$, and we denote by $\mathbf{c}$ a one hot vector corresponding to class $c$. We operate sequentially, assigning a label to each new data point we see using the following approximation of the true measure $\mu_K^q$, where $p(y_i = \mathbf{c} | y_1, \ldots, y_{i-1}) \approx \frac{1}{C} \exp(\frac{1}{T} \sum_{j<i} \mathbf{K}_{i,j} \mathbf{1}_{\{y_j = \mathbf{c}\}})$.

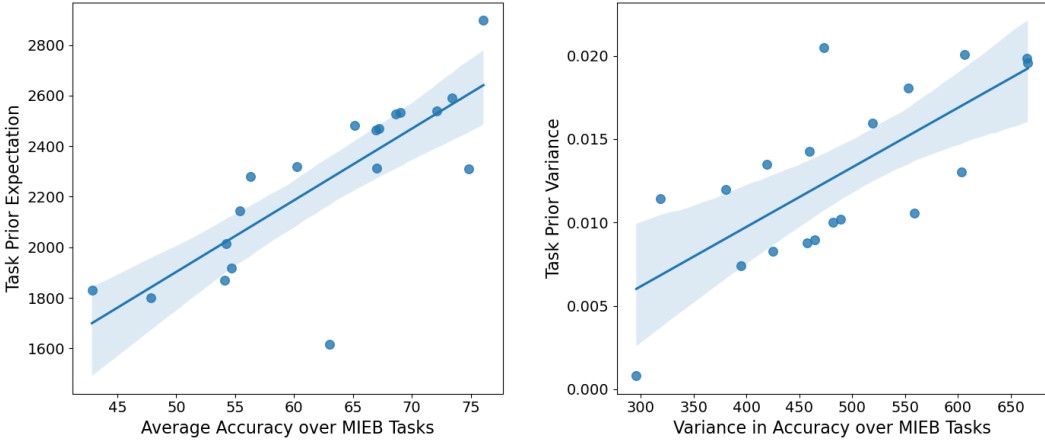

Figure 2: On Mini-ImageNet, task-prior estimates $E(K_G)$ and $\mathrm{Var}(E_G)$, computed from kernel matrices using the strongest model as the prior, correlate with the ground-truth mean and variance of linear-probe accuracies across tasks, using MIEB's image-classification models and tasks. We observe a correlation of $0.79$ between the Task Prior expectation and average accuracy, and a correlation of $0.71$ between the Task Prior variance and variance in accuracies.

We can then achieve an algorithmic speedup by using the factorization of our kernel matrix as $\mathbf{K} = \mathbf{Z}\mathbf{Z}^T$ (if we do not have access to the features, we can use for instance a Cholesky factorization here). Then we have,

$$\exp\left(\frac{1}{T}\sum_{j<i}\mathbf{K}_{i,j}\mathbf{1}_{\{y_j=\mathbf{c}\}}\right) = \exp\left(\frac{1}{T}\mathbf{Z}_i\sum_{j<i}\mathbf{Z}_j\mathbf{1}_{\{y_j=\mathbf{c}\}}\right). \tag{5}$$

From (5), we can devise our method for the sampling Algorithm 1. Using this algorithm, we are able to quickly sample labelings of the data points according to the Task Prior, as demonstrated in Figure 1.

## 3 Task Priors Predict Real Performance

We can use the equations in 2.3 as a way to evaluate model performance in a very fast way, i.e. without training any probes, or even assembling a collection of tasks / benchmarks. However, for this framework to be useful to practitioners, the performance on a hand curated collection of downstream tasks should follow the distribution implied by the Task Prior. In this section, we verify, for the hand-curated collection of downstream tasks for image classification found in MIEB[19], that our framework is able to predict downstream task performance.

We focus our work on a selection of the 19 models easily available through *huggingface*[18]. The Massive Image Embedding Benchmark (MIEB) paper [19] reports the accuracy of each of these models on each of 13 downstream classification tasks. Using the strongest model as our Task Prior, we find that the mean and variance of the linear probe accuracy across these downstream tasks correlate to $\mathbb{E}_{\mu_K}[\mathrm{Tr}(GK)]$ and $\mathrm{Var}(\mathrm{Tr}(GK))$ respectively, as demonstrated in Figure 2. We observe a strong correlation between the predictions made by our theory and the model performance as reported in MIEB.

Ultimately, our results show how one can utilize the Task Prior framework to predict model performance in a fast and easy way. We build on this work with further empirical evidence by evaluating the models found in *timm* [18], which we showcase in Appendix D.1.

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

# A Related Works

Many works aim to capture the performance of representation models primarily by intrinsic quantities about the model's features. For instance, RANKME measures the *effective rank* of the feature matrix and shows an empirical correlation with average linear-probe accuracy across several tasks [3]. LIDAR argues that a method built on Linear Discriminant Analysis serves as a proxy for downstream performance [16]. More recent works take a broader view, demonstrating that k-NN, few-shot fine-tuning, and clustering evaluations may all disagree in systematic ways [11]. Collectively, these studies show that properties intrinsic to the representation can forecast downstream success, but they still reduce performance to one or two scalar summaries.

A complementary line of work attacks the evaluation bottleneck by increasing the number of test tasks. In NLP, suites such as MTEB (56 embedding datasets) [12] and HELM (42 scenarios, seven axes of measurement) [9] provide broad coverage of downstream tasks. The same trend is exists in vision, with works such as *VideoEval* packaging twenty diverse video understanding datasets together [7], and frameworks such as MIEB [19] providing a curated collection of downstream tasks for vision and multi-modal models.. While these mega-benchmarks can be quite helpful for practitioners, they remain *finite* and expensive to create. Worse, even a hundred benchmarks sample only a vanishingly small corner of the large task space practitioners can care about.

Our *Task Priors* framework can be viewed as the missing bridge between these two threads. Like intrinsic metrics, it avoids needing a hand curated set of downstream targets, but like conglomerate benchmarks, it explicitly reasons about *many* tasks. Our framework echoes several well-known results from the classical theory of kernels. Notably, the trace term $\mathrm{Tr}(\mathbf{G}\mathbf{K})$ parallels the Hilbert–Schmidt Independence Criterion (HSIC) of Eq. (4) in [4]. Likewise, the term we get by taking the trace of $\mathbf{K}\mathbf{G}$ is precisely the same as "kernel alignment" studied in the context of generalization [1], obtained by flattening each matrix and taking their inner product, as $\langle \mathbf{K}, \mathbf{G} \rangle = \mathrm{Tr}(\mathbf{K}\mathbf{G})$.

There are some other related works that attack similar problems. In the computer vision space, VTAB [20] takes a similar distributional view of tasks, but does not precisely characterize the distribution of tasks. Similar to our derivations, [8] proposes a loss function based on the HSIC, which is an interesting avenue for future research.

## A.1 Limitations and Future Work

Despite these advances, several open issues remain. While the trace metrics correlate with probe accuracy, the correlation is not exact; closing this theory–practice gap will require a deeper empirical and theoretic study. Additionally, storing the full $n^2$ kernel is can be prohibitive for very large datasets, although the matrices we observe are highly structured; further leveraging sparsity and low-rank factorizations is an immediate direction for further work. Our analysis is domain-agnostic, but its effectiveness on understanding the representations of Large Language Models [14], and, more generally, in natural language processing remains to be demonstrated. Tackling these questions will not only sharpen the foundations introduced here but may also lead to AI systems that perform *consistently well* across the vast landscape of tasks encountered in practice.

 # B  Main Theoretical Results

 ## B.1  Proof of lemma 2.2

 *Proof.* Recall that $\mu(\mathbf{G}) = \frac{1}{Z_{T,\mathbf{K}}} e^{\frac{1}{T}\operatorname{Tr}(\mathbf{G}\mathbf{K})}$. Then we can compute that,

$$
\begin{aligned}
\mathbb{P}(\mathbf{G}_{i,j} = 1) &= \mathbb{E}_\mu[\mathbf{G}_{ij}] \\
&= \frac{1}{Z_{T,\mathbf{K}}} \sum_{\mathbf{G}:\mathbf{G}_{i,j}=1} e^{\frac{1}{T}\operatorname{Tr}(\mathbf{G}\mathbf{K})} \\
&= \frac{1}{Z_{T,\mathbf{K}}} \sum_{\mathbf{G}:\mathbf{G}_{i,j}=1} e^{\frac{1}{T}\sum_{i,j=1}^n \mathbf{G}_{i,j}\mathbf{K}_{i,j}} \\
&= \frac{1}{Z_{T,\mathbf{K}}} \sum_{\mathbf{G}:\mathbf{G}_{i,j}=1} \left[ \prod_{1\le k,l\le n} e^{\frac{1}{T}\mathbf{K}_{k,l}\mathbf{G}_{k,l}} \right].
\end{aligned}
$$

 Now notice we can let:

$$
w_1 = \sum_{\mathbf{G}:\mathbf{G}_{i,j}=1} \left[ \prod_{1\le k,l\le n} e^{\frac{1}{T}\mathbf{K}_{k,l}\mathbf{G}_{k,l}} \right],
$$

$$
w_0 = \sum_{\mathbf{G}:\mathbf{G}_{i,j}=0} \left[ \prod_{1\le k,l\le n} e^{\frac{1}{T}\mathbf{K}_{k,l}\mathbf{G}_{k,l}} \right].
$$

 And then,

$$
\mathbb{P}(\mathbf{G}_{i,j} = 1) = \frac{w_1}{w_0 + w_1}.
$$

 Notice though that we can write,

$$
\sum_{\mathbf{G}:\mathbf{G}_{i,j}=1} \left[ \prod_{1\le k,l\le n} e^{\frac{1}{T}\mathbf{K}_{k,l}\mathbf{G}_{k,l}} \right] = e^{\frac{1}{T}\mathbf{K}_{i,j}} \sum_{\mathbf{G}:\mathbf{G}_{i,j}=0} \left[ \prod_{1\le k,l\le n} e^{\frac{1}{T}\mathbf{K}_{k,l}\mathbf{G}_{k,l}} \right].
$$

 So,

$$
w_1 = e^{\frac{1}{T}\mathbf{K}_{i,j}} \cdot w_0.
$$

 Then we know that,

$$
\mathbb{P}(\mathbf{G}_{i,j} = 1) = \frac{e^{\frac{1}{T}\mathbf{K}_{i,j}} \cdot w_0}{w_0 + e^{\frac{1}{T}\mathbf{K}_{i,j}} \cdot w_0} = \frac{e^{\frac{\mathbf{K}_{i,j}}{T}}}{1 + e^{\frac{\mathbf{K}_{i,j}}{T}}} = \sigma\left(\frac{\mathbf{K}_{i,k}}{T}\right).
$$

 For the second part, we want to compute $\mathbb{P}(\mathbf{G}_{i,j}\mathbf{G}_{l,k} = 1)$, which we can note is clearly equivalent
 to $\mathbb{P}(\mathbf{G}_{i,j} = 1 \ \wedge \ \mathbf{G}_{l,k} = 1)$. As before, we can compute,

$$
\begin{aligned}
\mathbb{P}(\mathbf{G}_{i,j}\mathbf{G}_{l,k} = 1) &= \mathbb{E}_\mu[\mathbf{G}_{i,j}\mathbf{G}_{l,k}] \\
&= \frac{1}{Z_{T,\mathbf{K}}} \sum_{\mathbf{G}:\mathbf{G}_{i,j}\mathbf{G}_{l,k}=1} e^{\frac{1}{T}\operatorname{Tr}(\mathbf{G}\mathbf{K})} \\
&= \frac{1}{Z_{T,\mathbf{K}}} \sum_{\mathbf{G}:\mathbf{G}_{i,j}\mathbf{G}_{l,k}=1} e^{\frac{1}{T}\sum_{i,j=1}^n \mathbf{G}_{i,j}\mathbf{K}_{i,j}} \\
&= \frac{1}{Z_{T,\mathbf{K}}} \sum_{\mathbf{G}:\mathbf{G}_{i,j}\mathbf{G}_{l,k}=1} \left[ \prod_{1\le k,l\le n} e^{\frac{1}{T}\mathbf{K}_{k,l}\mathbf{G}_{k,l}} \right].
\end{aligned}
$$

 We then let,

$$
w_1 = \sum_{\mathbf{G}:\mathbf{G}_{i,j}\mathbf{G}_{l,k}=1} \left[ \prod_{1\le k,l\le n} e^{\frac{1}{T}\mathbf{K}_{k,l}\mathbf{G}_{k,l}} \right],
$$

242

$$w_0 = \sum_{\mathbf{G}:\mathbf{G}_{i,j}\mathbf{G}_{l,k}=0} \left[ \prod_{1\le k,l\le n} e^{\frac{1}{T}\mathbf{K}_{k,l}\mathbf{G}_{k,l}} \right].$$

243 Which, we can note if $(i,j) \ne (l,k)$, then we have:

$$w_0 = \sum_{\mathbf{G}:\mathbf{G}_{i,j}\mathbf{G}_{l,k}=0} \left[ \prod_{1\le k,l\le n} e^{\frac{1}{T}\mathbf{K}_{k,l}\mathbf{G}_{k,l}} \right]$$

$$= \sum_{\mathbf{G}:\mathbf{G}_{i,j}=0,\mathbf{G}_{l,k}=0} \left[ \prod_{1\le k,l\le n} e^{\frac{1}{T}\mathbf{K}_{k,l}\mathbf{G}_{k,l}} \right] + \sum_{\mathbf{G}:\mathbf{G}_{i,j}=0,\mathbf{G}_{l,k}=1} \left[ \prod_{1\le k,l\le n} e^{\frac{1}{T}\mathbf{K}_{k,l}\mathbf{G}_{k,l}} \right]$$

$$+ \sum_{\mathbf{G}:\mathbf{G}_{i,j}=1,\mathbf{G}_{l,k}=0} \left[ \prod_{1\le k,l\le n} e^{\frac{1}{T}\mathbf{K}_{k,l}\mathbf{G}_{k,l}} \right]$$

$$= (e^{\frac{1}{T}(\mathbf{K}_{i,j}+\mathbf{K}_{l,k})})^{-1} \sum_{\mathbf{G}:\mathbf{G}_{i,j}=1,\mathbf{G}_{l,k}=1} \left[ \prod_{1\le k,l\le n} e^{\frac{1}{T}\mathbf{K}_{k,l},\mathbf{G}_{k,l}} \right]$$

$$+ (e^{\frac{1}{T}\mathbf{K}_{i,j}})^{-1} \sum_{\mathbf{G}:\mathbf{G}_{i,j}=1,\mathbf{G}_{l,k}=1} \left[ \prod_{1\le k,l\le n} e^{\frac{1}{T}\mathbf{K}_{k,l}\mathbf{G}_{k,l}} \right]$$

$$+ (e^{\frac{1}{T}\mathbf{K}_{l,k}})^{-1} \sum_{\mathbf{G}:\mathbf{G}_{i,j}=1,\mathbf{G}_{l,k}=1} \left[ \prod_{1\le k,l\le n} e^{\frac{1}{T}\mathbf{K}_{k,l}\mathbf{G}_{k,l}} \right]$$

$$= (e^{-\frac{1}{T}(\mathbf{K}_{i,j}+\mathbf{K}_{l,k})} + e^{-\frac{1}{T}\mathbf{K}_{i,j}} + e^{-\frac{1}{T}\mathbf{K}_{l,k}}) \sum_{\mathbf{G}:\mathbf{G}_{i,j}=1,\mathbf{G}_{l,k}=1} \left[ \prod_{1\le k,l\le n} e^{\frac{1}{T}\mathbf{K}_{k,l}\mathbf{G}_{k,l}} \right]$$

$$= (e^{-\frac{1}{T}(\mathbf{K}_{i,j}+\mathbf{K}_{l,k})} + e^{-\frac{1}{T}\mathbf{K}_{i,j}} + e^{-\frac{1}{T}\mathbf{K}_{l,k}}) w_1.$$

244 Then we can write that,

$$w_0 + w_1 = (1 + e^{-\frac{1}{T}(\mathbf{K}_{i,j}+\mathbf{K}_{l,k})} + e^{-\frac{1}{T}\mathbf{K}_{i,j}} + e^{-\frac{1}{T}\mathbf{K}_{l,k}}) w_1.$$

245 So then,

$$\mathbb{P}(\mathbf{G}_{i,j}\mathbf{G}_{l,k} = 1) = \frac{w_1}{w_0 + w_1}$$

$$= \frac{1}{(1 + e^{-\frac{1}{T}(\mathbf{K}_{i,j}+\mathbf{K}_{l,k})} + e^{-\frac{1}{T}\mathbf{K}_{i,j}} + e^{-\frac{1}{T}\mathbf{K}_{l,k}})}$$

$$= \frac{1}{(1 + e^{-\frac{1}{T}\mathbf{K}_{i,j}})(1 + e^{-\frac{1}{T}\mathbf{K}_{l,k}})}$$

$$= \sigma(\frac{\mathbf{K}_{i,j}}{T})\sigma(\frac{\mathbf{K}_{l,k}}{T}).$$

246 □

## B.2 Proof of theorem 2.3

248 *Proof.* The first equality in the equation follows from the linearity of expectation, and the characteri-
249 zation that,

$$\mathrm{Tr}(\mathbf{M}\mathbf{G}) = \sum_{1\le i,j\le n} \mathbf{M}_{i,j}\mathbf{G}_{i,j},$$

250 for $\mathbf{M}, \mathbf{G}$ symmetric matrices. Then, notice that this is a weighted sum of independent Bernoulli
251 random variables. So, $\mathbb{E}_{\mathbf{G}\sim\mu_{\mathbf{K}}}[\mathbf{G}_{i,j}] = \mathbb{P}_{\mathbf{G}\sim\mu_{\mathbf{K}}}(\mathbf{G}_{i,j} = 1)$ and we can apply the above lemma and
252 we are done.

For the second part, since this is a sum of independent random variables, we may use,

$$\text{Var}(\text{Tr}(\mathbf{M}\mathbf{G})) = \sum_{1 \le i,j \le n} \mathbf{M}_{i,j}^2 \, \text{Var}(G_{i,j}) \tag{6}$$

$$= \sum_{1 \le i,j \le n} \mathbf{M}_{i,j}^2 \mathbb{P}(\mathbf{G}_{i,j} = 1)(1 - \mathbb{P}(\mathbf{G}_{i,j} = 1)) \tag{7}$$

$$= \sum_{1 \le i,j \le n} \mathbf{M}_{i,j}^2 \sigma(\mathbf{K}_{i,j}/T)(1 - \sigma(\mathbf{K}_{i,j}/T)). \tag{8}$$

$\square$

## C  Sampling Algorithm

---
**Algorithm 1** Prefix Sampler for Multi-Class Task Prior

---
**Require:** $\mathbf{Z} \in \mathbb{R}^{n \times r}$                  $\triangleright$ factor so $\mathbf{K} \approx \mathbf{Z}\mathbf{Z}^\top$
**Require:** $T > 0$                      $\triangleright$ temperature
**Require:** $q \in \mathbb{N}$                $\triangleright$ number of classes
**Ensure:** $labels \in \{0, \dots, q-1\}^n$
  1: allocate $labels[1{:}n]$
  2: $\mathbf{U} \leftarrow 0_{r \times q}$                $\triangleright$ class-wise prefix sums
  3: **for** $i = 1 \dots n$ **do**
  4:      $h \leftarrow \left(\frac{1}{T}\right)(\mathbf{Z}[i,:]\,\mathbf{U})$        $\triangleright$ length $q$ vector
  5:      $h \leftarrow h - \max(h)$             $\triangleright$ stabilize
  6:      $p \leftarrow \exp(h); \quad p \leftarrow p/\sum p$
  7:      $c \sim \text{CategoricalSample}(p)$
  8:      $labels[i] \leftarrow c$
  9:      $\mathbf{U}[:,c] \mathrel{+}= \mathbf{Z}_{i,:}$
10: **end for**
11: **return** $labels$

---

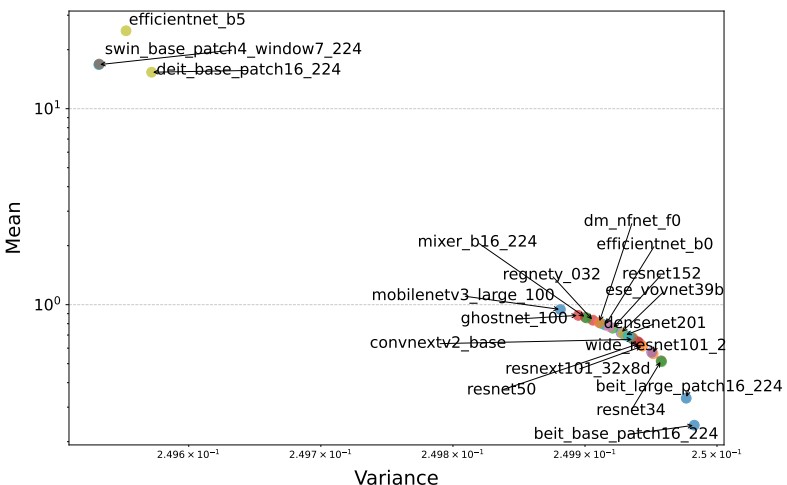

Figure 4: Here we plot the expectation and variance of $\mathrm{Tr}(GM)$, where $M$ is the centered cosine similarity kernel matrix for each models features generated from mini-imagenet[13], where the expectation is taken against $\mu_K$. Here we use $K$ as the centered cosine similarity kernel matrix for *efficientnet_b5*. Please see the appendix for more information and ablation on temperature and choice of kernel.

## D    Additional Empirical Studies

### D.1    Linear Probe Performance

We can use the equations in 2.3 as a way to evaluate model performance in a very fast way, i.e. without training any probes, or even assembling a collection of tasks / benchmarks. We demonstrate this in Figure 4 on a selection of models from *timm* and on a subset of $8, 192$ images from *mini-imagenet* [13]. We use the centered cosine similarity as the choice of kernel matrix here and in the rest of the experiments in this paper. We find that the mean and variance are negatively correlated, implying that models that perform well on average tend also to perform better across a variety of tasks. From the selection of models we tested, we find that *efficientnet* [15] performs the best, even beating more modern vision transformers [2].

A central claim of this paper is that the two kernel statistics, $\mathbb{E}_{\mu_K}[\mathrm{Tr}(GK)]$ and $\mathrm{Var}_{\mu_K}(\mathrm{Tr}(GK))$, can predict a representation's downstream performance as measured by linear probes. Using our sampling algorithm, we draw tasks from a prior $\mu_K$ induced by *efficientnet_b5*. For each of 33 models from *timm* [18], we train an independent linear probe on every sampled task with *efficientnet_b5* and record the resulting accuracies. We then compare this to simply computing $\mathbb{E}_{\mu_K}[\mathrm{Tr}(GK)]$ and $\mathrm{Var}(\mathrm{Tr}(GK))$ as per 2.3.

We report the results of this study in Figure 5, where we find that models that perform better on average also tend to have a better variance over tasks. This is a finding we will corroborate by directly measuring $\mathbb{E}_{\mu_K}[\mathrm{Tr}(GK)]$ and $\mathrm{Var}(\mathrm{Tr}(GK))$ in Figure 5.

We note that the expectation and variance of $\mathrm{Tr}(GK)$ as shown in Figure 4, tends to exhibit the same trends as the models linear probe performance on sampled tasks, as seen in Figure 5, where stronger models tend to have a higher average accuracy / trace, as well as a lower variance.

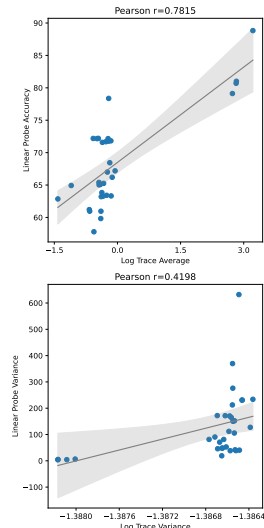

Figure 3: Correlation between the mean and variance of $\mathrm{Tr}(GK)$ and the performance of linear probes on the same representations, with $95\%$ confidence intervals.

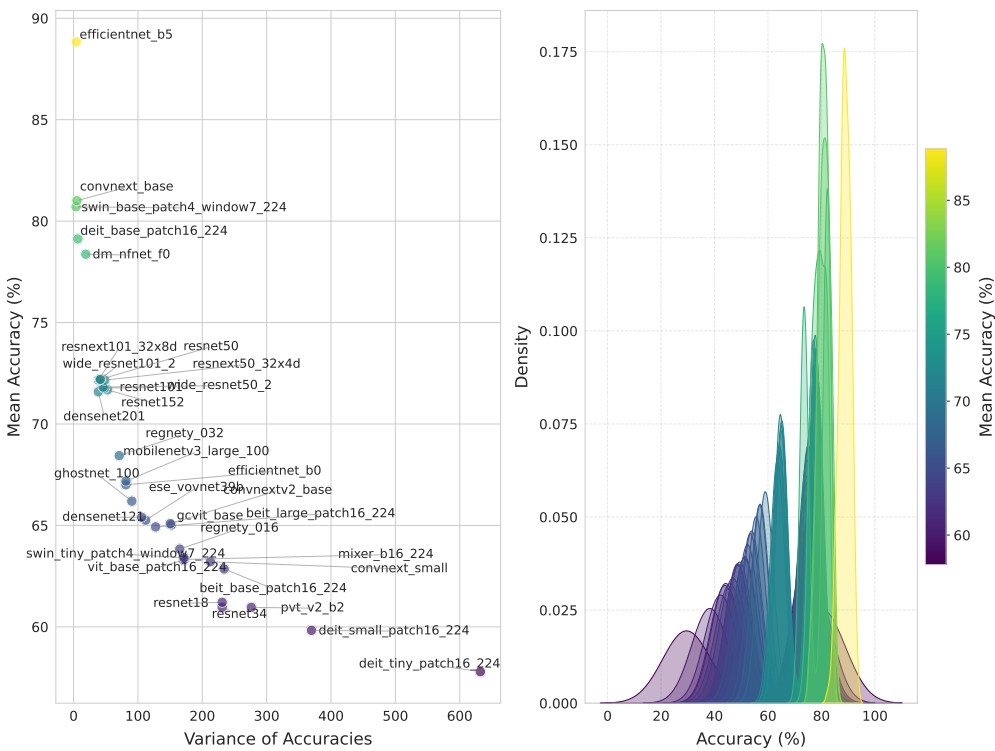

Figure 5: Linear probe performance of a selection of models from *timm* on a distribution of binary labels sampled by the *Task Prior* on the mini-Imagenet dataset. Here we use *efficientnet_b5* as the backbone model for the Task Prior distribution.

## D.2 Ablation on Choice of Kernel

In Figure 6, we can see how the choice of Task Prior kernel matrix affects the downstream computation of the mean and variance of $\mathrm{Tr}(\mathbf{MG})$. As we might expect, we see that generally the mean $\mathbb{E}_{\mathbf{G} \sim \mu_{\mathbf{K}}}[\mathrm{Tr}(\mathbf{MG})]$ is higher when the Task Prior kernel $\mathbf{K}$ matrix is the same as the matrix being evaluated $\mathbf{M}$. We don't see this same behavior with the variance $\mathrm{Var}_{\mathbf{G} \sim \mu_{\mathbf{K}}}(\mathrm{Tr}(\mathbf{MG}))$.

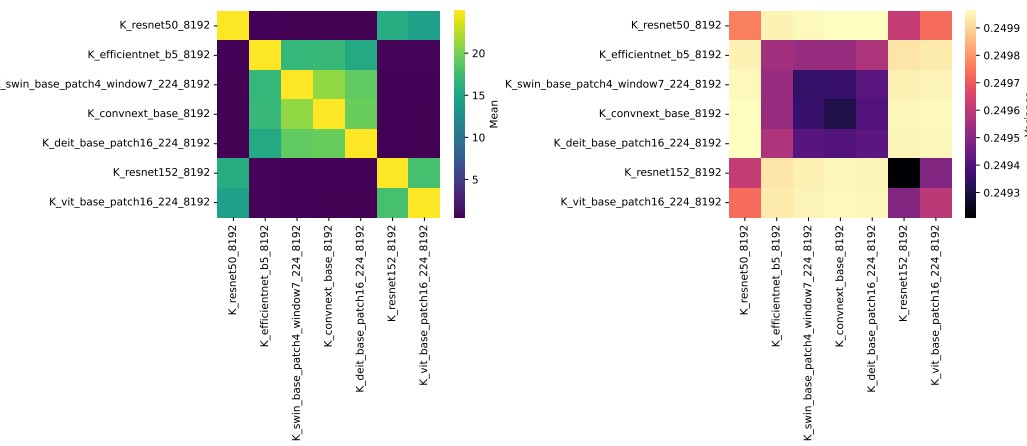

Figure 6: Here, we show a comparison of how the choice of Task Prior kernel $\mathbf{K}$, reflected here in the color of the data points, affects the the evaluation of the mean and variance of $\mathrm{Tr}(\mathbf{MG})$. Each point is computed via the exact formulas given in 2.3, with a temperature of $T = 0.01$.

### D.3  Ablation on the Temperature Parameter

In Figure 7, we can see the effect of the sampler changing the temperature in the measure. We can see how increasing temperature increases diversity but also brings us closer to a uniform distribution over labels.

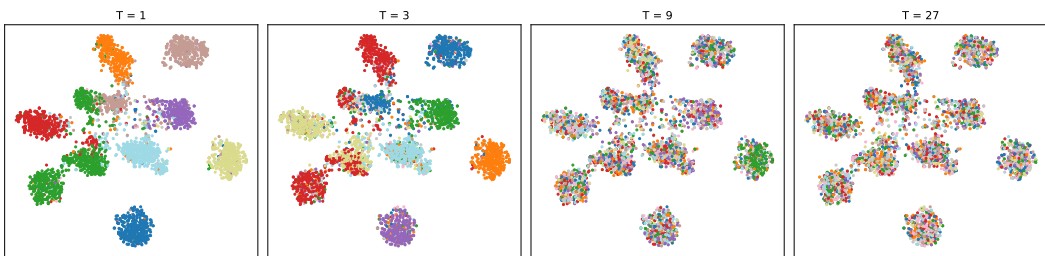

Figure 7: Here we show a TSNE plot of Imagenette, with labels generated by the sampling Algorithm 1 for four choices of temperature.

