# OpenReview forum: "Task Priors: Enhancing Model Evaluation by Considering the Entire Space of Downstream Tasks"
_NeurIPS.cc/2025/Workshop/UniReps — UniReps2025_

### Official Review · Reviewer_5pgN · 2025-09-06
**Review of Task Priors**

**Confidence:** 4

**Review:**

### **Summary**
The paper tackles a bottleneck in model evaluation which is the reliance on fixed benchmark suites that only cover a narrow slice of possible downstream tasks. The authors introduce Task Priors, a probabilistic distribution over downstream tasks, allowing one to estimate a model’s expected performance without training probes or collecting new benchmarks. They provide theoretical derivations and empirical validation on image embedding models on MIEB.

### **Strengths**
- The motivation is strong and focuses on an important problem in the current AI evaluation practices.
- It has strong theoretical soundness and provides clear derivations for their study.

### **Weaknesses**
- Results are entirely on vision embedding models with MIEB. No evidence is provided that this generalizes to NLP or other domains.
- While the correlations are encouraging, they are not strong enough to convincingly replace real downstream evaluation. Results on other benchmarks can strengthen their claim.
- The strong model is chosen as the prior which affects the results. Robustness to the choice of prior must be explored.
- Presentation issues
  - Line 18 has a citation to the repository `rwightman/pytorch-image-models` while mentioning download statistics of various models on HuggingFace.
  - Line 109 again cites the repository `rwightman/pytorch-image-models` while mentioning HuggingFace.
  - Figure 4 mentions "Please see the appendix for more information" while already being in the appendix.
  - Figures 4 and 5 are not readable and provide a clustered model names. A better visualization is needed.
  - Line 9 has a missing full stop.

### **Questions**
- Could the authors elaborate on the criteria for selecting the prior kernel K? Is there a theoretical justification or empirical guideline for choosing the "strongest model" as the prior, or are there other valid choices? How robust are the results to variations in this choice?

### **Conclusion**
Although the paper provides a novel and important method for evaluation, the paper seems like a rushed submission and requires rectification of the formatting and grammatical errors.

**Score:**

3

**Topic Fit:**

3

---

### Official Review · Reviewer_dVVA · 2025-09-15
**fresh ideas and perspective, intuitive math, but clarity and presentation quality can be much improved**

**Confidence:** 3

**Review:**

Summary

This paper introduces Task priors as a new framework for ML model evaluation, providing an alternative to the current approach, which relies on a fixed collection of hand-picked downstream benchmarks.



Strengths

1. To the best of my knowledge, this seems like very original work with fresh ideas that would be of interest to a very wide audience. Furthermore, it looks like there is plenty of potential for further development in follow-up studies.

2. I found the math straightforward and easy to follow, and it seems sound.

3. The numerics in the appendix seem thorough.

Weaknesses

1. I found some points to be unclear or confusing:

(a) Figure 1 - the standard labels in panel (a) seem to be almost the only sensible way to assign labels, with very little wiggle room. Hence, the various labelings shown in panel (b) are either very close to those in (a) or seem to make very little sense. Did I miss something here? Perhaps other datasets admit a wide range of sensible labelings and hence would be more fitting.

(b) Strongest model - this term is used several times and I could not find a definition (I may have missed this). Does this just mean maximum mean and minimum variance? Please clarify.

2. I found the structure of the text a little discontinuous, especially in the discussion and implications section, which I think can be much improved.

Overall, I found the ideas put forth in this paper to be quite original and interesting, with rather straightforward math, and I believe it may be of interest to quite a wide audience and thus should be accepted, hence the score of 4. However, I did raise some concerns regarding the clarity of the discussion and implications of this work, the definition of some main concept,s and the main messages in some of the figures. I hope these issues will be addressed by the authors.

**Score:**

4

**Topic Fit:**

3

---

### Official Review · Reviewer_qehj · 2025-09-15
**Review of "Task Priors"**

**Confidence:** 3

**Review:**

The paper proposes a kernel-induced Gibbs measure over label graphs G to define a distribution of downstream classification tasks. With this “Task Prior”, the expectation/variance of alignment scores with another kernel M are computed in closed form. A fast approximate sampler is given to draw concrete labelings. Correlations with linear-probe accuracies on Mini-ImageNet/MIEB are reported.

Strengths:
- Benchmark-centric evaluation is indeed a bottleneck. A distributional view of tasks is thought-provoking.
- The trace alignment/Gibbs construction is elegant and connects to kernel alignment. Expectations/variances have closed forms.
- The included Colab notebook is appreciated, this improves accessibility.

Weaknesses:
- Despite the framing, the method models classification labelings via G. It does not cover other major task families (retrieval/ranking, detection/segmentation, generative tasks, calibration, robustness), so “entire space of downstream tasks” reads as overclaiming. Please scope the claim to classification label graphs and discuss extensions.
- The strongest empirical claim uses the “strongest model” as the prior K, then evaluates other models M. This can favor models whose kernels resemble the chosen prior; indeed, you also show the mean increases when M=K. This introduces bias.
- The paper says the stats can be computed “in milliseconds,” yet the dominant cost can be forming/centering K,M ( O(nd) for features + O(n^2) memory). Please provide wall-clock and memory scaling.
- The abstract’s language (‘grand goal’) feels overstated; please use neutral, scoped phrasing that reflects the paper’s actual contribution.
- In Lemma 2.2, Eq. (3) appears to reuse the same indices; I believe the second term should be $G_{\ell,k}$. Please correct to avoid confusion.
- Key limitations are only in Appendix A.1; please surface a concise limitations paragraph in the main text.
- Fig. 3 reports Pearson r, but other correlations (0.79/0.71) are not identified as Pearson or Spearman. Please standardize the reporting of correlation type throughout. It would also be valuable to provide both Pearson and Spearman coefficients, since rank-based correlation is particularly relevant for comparing model performance.

This is a thought-provoking submission that opens a novel direction in evaluation methodology. With clearer exposition and broader experiments, it could be more impactful.

**Score:**

2

**Topic Fit:**

3